# Synthesis, Characterization, and the Antioxidant Activity of Carboxymethyl Chitosan Derivatives Containing Thiourea Salts

**DOI:** 10.3390/polym11111810

**Published:** 2019-11-04

**Authors:** Xueqi Sun, Jingjing Zhang, Yuan Chen, Yingqi Mi, Wenqiang Tan, Qing Li, Fang Dong, Zhanyong Guo

**Affiliations:** 1Key Laboratory of Coastal Biology and Bioresource Utilization, Yantai Institute of Coastal Zone Research, Chinese Academy of Sciences, Yantai 264003, China; xqsun@yic.ac.cn (X.S.); jingjingzhang@yic.ac.cn (J.Z.); yuanchen@yic.ac.cn (Y.C.); yqmi@yic.ac.cn (Y.M.); wqtan@yic.ac.cn (W.T.); qli@yic.ac.cn (Q.L.); 2Center for Ocean Mega-Science, Chinese Academy of Sciences, 7 Nanhai Road, Qingdao 266071, China; 3University of Chinese Academy of Sciences, Beijing 100049, China

**Keywords:** antioxidant ability, characterization, chitosan derivatives, thiourea salts

## Abstract

A new class of chitosan derivatives possessing thiourea salts were synthesized to improve the solubility and the antioxidant activity of chitosan. Firstly, chitosan was modified to carboxymethyl chitosan, combining carboxymethyl chitosan with thiourea salts that have different structures to form new chitosan derivatives. The chitosan and chitosan derivatives were characterized by FT-IR, ^13^C NMR, TGA, and elemental analyses. The new peaks of thiourea salts could be clearly observed at about 1240 cm^–1^ in the IR spectra, and the peak of C=S was clearly observed at around 180 ppm in the ^13^C NMR. IR spectra and ^13^C NMR of the structural units of these polymers validated the chitosan derivatives possessing thiourea salts were successfully synthesized. Their antioxidant properties were tested, including DPPH-radical scavenging ability, superoxide-radical scavenging ability, and hydroxyl-radical scavenging ability. Our results suggested the increase of thiourea salt groups in chitosan derivatives promotes the scavenging effect. The scavenging activity of 4TMCMCS, TMCMCS. 4,4DCMCS, and 4ACMCS against DPPH-radical and superoxide-radical were more than 90% at 1.6 mg/mL, respectively. In the cytotoxicity assay, no cytotoxicity was observed for the L929 cells with chitosan and its derivatives at all testing concentrations. These results demonstrated that the combination of chitosan and thiourea salt groups improved the antioxidant activity of chitosan, and the antioxidants or free radical scavengers based on natural polymers and thiourea salts showed potential applications.

## 1. Introduction

Oxygen is involved in all life activities of plants. During the metabolic process, 4%–5% oxygen forms superoxide anions, and most of the oxygen forms water by combining with hydrogen. Water produced can form hydrogen peroxide by metabolism, which consists of free radicals. The oxygen radicals will be formed, because the oxygen is the easiest way to get electrons in the living body. These free radicals will attack membranes such as cell membranes. Because they are unstable, they have high oxidative activity and their activity is lively. Lipid peroxidation increases when they react with unsaturated fatty acids, which are present in the cell membrane [1]. The chain reaction of free radicals will occur because lipid peroxidation may decompose more free radicals. At this rate, the integrity of the membrane structure will be completely destroyed.

Although all aerobic organisms can produce reactive oxygen species, there is also a complete active oxygen scavenging system (antioxidant enzymes and antioxidants) in the organisms which can convert active oxygen into less active substances. The basal body is subjected to protection [2,3]. However, the scavenging ability of free radicals decreases with age, and sometimes oxygen free radicals cannot be completely cleared away. In the course of growth, plants will suffer from stresses, such as pathogen infection, extreme temperatures, drought and flood, metal ions, chemical agents and so on [4,5]. When multiple stressors occur, if the active oxygen is not removed in time, a large amount of active oxygen will be produced in the cells. This will be converted into free radicals under certain conditions, also causing membrane lipid peroxide and the membrane permeability loss. This will cause a series of physiological and biochemical changes, including metabolic disorders [6]. As long as one of the stressors breaks down, the plants will be targeted. Therefore, the discovery of new synthetic antioxidants is important in the fields of organic transformations and antioxidant synthesis that protect the plants from the damage of aging and other stressors.

The thiourea derivative is a kind of thiocarbonyl compound which can exist stably, and it has a wide range of compounds. When it is used in the fields of medicine, chemistry, agronomy and so on, its application prospects are broad, and thus it has become a focus of scholars at home and abroad. Thiocarbazone, the important member of thiourea derivatives, has a wide range of physiological pharmacological activities. It is a compound of the structure Figure 1, in which an amino group is condensed with a ketone or an aldehyde carbonyl group to form a thiocarbazone [7]. It plays an outstanding role in DPPH free radical scavenging, anti-dyslipidemia, and scavenging of oxygen free radical [8,9,10]. The thiosemicarbazone is considered as one of the most important scaffolds and is embedded in many biologically-active compounds. It arouses wide public interest in terms of anti-cancer, anti-bacterial, anti-oxidation, bactericidal and insecticidal properties, regulating plant physiological functions [11,12].

A large number of studies have shown that bioactive polysaccharides of marine plants have a strong ability to scavenge free radicals. Chitosan is one of them, and it is derived from the most abundant chitin, which occurs in fungal and yeast cell walls in krill, lobster and crab tendons and shells, and shrimp shells, as well as in insect cuticle. It has become a research hotspot because of its non-toxic, non-polluting and biodegradable properties [13]. Since chitosan is only soluble in acidic aqueous media, this limits its enormous value in many applications (food, cosmetics, biomedical, and pharmaceutical applications) [14]. Chitosan has a unique structure, containing active amino and hydroxyl groups, on which other active groups can be introduced to improve its solubility and biological activity and generate new active functions [15]. Carboxymethylation is one of the important means used to modify polymer. For example, carboxycelluloses are important derivatives of natural cellulose polymers, and they have been widely used in many biomedical, agricultural, and wastewater treatment field applications [16,17,18]. Even though thiocarbazone has strong biological activity, it is toxic. Therefore, by modifying chitosan to carboxymethyl chitosan, we will combine carboxymethyl chitosan with thiosemicarbazone derivatives to form new chitosan derivatives. Through such a method, the toxicity of the thiocarbazone derivative is reduced, and the solubility and biological activity of chitosan are improved. We briefly describe the chemical modifications of chitosan, an area in which the syntheses have been proposed tentatively, but are not yet developed on an industrial scale. This report emphasizes recent papers on the high value-added applications of these drugs in plant growth regulators.

## 2. Materials and Methods

### 2.1. Materials

Chitosan was purchased from Golden-Shell Pharmaceutical Co. Ltd. (Zhejiang, China). Its degree of deacetylation is 73.5% and the viscosity-average molecular weight is 5–8 kDa. Thiocarbamides (thiosemicarbazone, 4-methylthiosemicarbazide, 4,4-dimethyl-3-thiosemicarbazide, 4-amino-3-indolyl-5-mercapto-1,2,4 triazole) were purchased from Sinopharm Chemical Reagent Co., Ltd. (Shanghai, China), and chloroacetic acid was purchased from Sigma-Aldrich Chemical Corp. (Shanghai, China). The other reagents such as ethanol, dimethyl carbinol, and sodium hydroxide are analytical grade and were supplied by Sinopharm Chemical Reagent Co., Ltd., Shanghai, China.

### 2.2. Analytical Methods

#### 2.2.1. Fourier Transform Infrared (FT-IR) Spectroscopy

FT-IR spectrometers, in the range of 4000–400 cm^–1^ with resolution of 4.0 cm^–1^, were recorded on a Jasco-4100 (Tokyo, Japan, provided by JASCO China (Shanghai) Co. Ltd., Shanghai, China). All samples were ground and mixed with KBr disks for testing.

#### 2.2.2. Nuclear Magnetic Resonance (NMR) Spectroscopy

^13^C Nuclear magnetic resonance (^13^C NMR) spectra were recorded on a Bruker AVIII-500 Spectrometer (Switzerland, provided by Bruker Tech. and Serv. Co., Ltd. Beijing, China) at 25 °C with tetramethyl silane (TMS) as internal standard on ppm scale. For analyses, 50 mg of samples were dissolved in 1.0 mL of 99.9% Deuterium Oxide (D_2_O).

#### 2.2.3. Elemental Analyses

The extent of functionalization (degree of substitution) in chitosan derivatives was quantitatively assessed using the elemental analyses of burnable. The detection of elemental carbon, hydrogen, nitrogen and sulfur in chitosan derivatives was performed on a Vario EL III (Elementar, Germany). The degrees of substitution (DS) of chitosan derivatives were calculated on the basis of the percentages of carbon and nitrogen or carbon and sulfur according to the following equations [19]:(1)$$DS_{1}=\frac{n_{1}\times \;M_{C}-M_{N}\times \;W_{C/N}}{n_{2}\times \;M_{C}}$$

(2)$$DS_{2}=\frac{M_{N}\times \;W_{C/N}+n_{2}\times \;M_{C}\times DS_{1}-n_{1}\times \;M_{C}}{n_{3}\times \;M_{C}}$$

(3)$$DS_{3}=\frac{\left(n_{1}-n_{2}\times DS_{1}+n_{3}\times DS_{2}\right)\times \;M_{C}\;}{\;M_{C/S}\times \;m_{1}\times \;M_{S}-n_{4}\times \;M_{C}}$$

(4)$$DS_{4}=\frac{\left(n_{1}-n_{2}\times DS_{1}+n_{3}\times DS_{2}\right)\times \;M_{C}}{\;M_{C/S}\times \;m_{1}\times \;M_{S}-n_{5}\times \;M_{C}}$$

(5)$$DS_{5}=\frac{\left(n_{1}-n_{2}\times DS_{1}+n_{3}\times DS_{2}\right)\times \;M_{C}}{M_{C/S}\times m_{1}\times \;M_{S}-n_{6}\times \;M_{C}}$$

(6)$$DS_{6}=\frac{\left(n_{1}-n_{2}\times DS_{1}+n_{3}\times DS_{2}\right)\times \;M_{C}\;}{\;M_{C/S}\times \;m_{1}\times \;M_{S}-n_{7}\times \;M_{C}}$$

*M*_C_, *M*_N_ and *M*_S_ are the molar mass of carbon, nitrogen and sulfur, *M*_C_ = 12, *M*_N_ = 14, *M*_S_ = 32; n_1_, *n*_2_, *n*_3_, *n*_4_, *n*_5_, *n*_6_, and n_7_ are the number of carbon of chitin, two carboxymethyl group, two thiocarbamides (thiosemicarbazone, 4-methylthiosemicarbazide, 4,4-dimethyl-3-thiosemicarbazide, 4-amino-3-indolyl-5-mercapto-1,2,4 triazole), *n*_1_ = 8, *n*_2_ = 2, *n*_3_ = 4, *n*_4_ = 4, *n*_5_ =2, *n*_6_=6, *n*_7_=4, W_C/N_ represents the mass ratio between carbon and nitrogen, and W_C/S_ represents the mass ratio between carbon and sulfur. 

#### 2.2.4. Thermogravimetric Analysis (TGA)

The thermogravimetric analysis (TGA) was done on the TGA/DSC1/1100 (Mettler-Toledo). The samples were heated from room temperature 25 °C–600 °C with a temperature rate of change of 10 °C per min. The thermogravimetric analysis was performed in N_2_.

### 2.3. Synthesis of Chitosan Derivatives

#### 2.3.1. Synthesis of N, O-carboxymethyl Chitosan (N, O-CMCS)

Chitosan (1.62 g, 10 mmol) was stirred vigorously in 20 mL of isopropanol-water system (volume ratio is 4:1) at 50 °C, then sodium hydroxide (1.2 g, 30 mmol) was added. After stirring for 1 h at 50 °C, chloroacetic acid (4.72 g, 50 mmol) was dissolved in 6ml isopropanol, isopropyl chloroacetate solution was then dropped into the chitosan isopropyl alcohol solution slowly. The solution was heated at 50 °C and stirred for 4 h. The product was isolated and washed by pouring the reaction solution into absolute ethanol solution (200 mL) with 70%, 80%, 90% content successively. The precipitate was collected by filtration, washed with ethanol and *N, O-*CMCS was obtained by freeze-drying overnight in a vacuum. Yields: 75.85% [20,21].

#### 2.3.2. Synthesis of 4MTCMCS, TCMCS, 4,4DCMCS, and 4ACMCS

A certain amount of thiosemicarbazone (0.876 g, 12 mmol), 4-methylthiosemicarbazide (1.26 g, 12 mmol), 4,4-dimethyl-3-thiosemicarbazide (1.428 g, 12 mmol), and 4-amino-3-indolyl-5-mercapto-1,2,4 triazole (1.752 g, 12 mmol) was dissolved in 1.2 mol/L hydrochloric acid solution. The volume of hydrochloric acid solution was 10 mL, and it was stirred for 30 min at room temperature. Upon obtaining the thiourea salts that we needed, *N, O-*carboxymethyl chitosan (1.2 g, 2.8 mmol) was dissolved in the thiourea salt solution prepared above, mixed with the above-mentioned thiourea salt, and stirred for 12 h at room temperature. Then, the reacted solution was poured into 100 mL of absolute ethyl alcohol to precipitate the product. The precipitate was filtered and washed carefully three times with absolute ethyl alcohol, and extracted in a Soxhlet apparatus with ethanol 48 h [22,23]. Then the products were dissolved in water and subjected to dialysis with distilled water for 48 h in a dialysis bag with a molecular weight cut-off of 100 g/mol (molecular weight cut off). The products were obtained by freeze-drying. Yields:4MTCMCS 42.56%, TCMCS 27.71%, 4,4DCMCS 26.52%, and 4ACMCS 20.41%. (Scheme 1.)

### 2.4. Antioxidant Activity Assay

#### 2.4.1. DPPH Radicals’ Scavenging Ability Assay

The DPPH-radical scavenging capacity of the products was evaluated using some researchers’ methods [24]. The method is as follows: The stock solution was prepared by dissolving 35.49 mg DPPH with 500 mL ethyl alcohol and then stored in a dark place. Chitosan and its derivatives with 1 mL concentration of 0.1, 0.2, 0.4, 0.8, and 1.6 mg/mL were added to the test tube (use isovolumetric distilled water as contrast). The reaction mixture, a total volume of 3.0 mL, containing 1 mL sample solution with concentration of 0.1, 0.2, 0.4, 0.8, and 1.6 mg/mL, was incubated with DPPH (2 mL) at 25 °C for 20 min in dark place. Control group: 2 mL DPPH took the place of the 2 mL ethyl alcohol. Vc (Vitamin C) was used as a positive control. The ultraviolet absorbance of the mixture at 517 nm was determined by the microplate reader. Three replicates for each sample were tested. The scavenging rate of the DPPH-radical was calculated according to the following equation:(7)$$Scavenging\;index\;\left(\%\right)=\left[1-\frac{A_{\;sample\;517nm}-A{\;}_{control\;517nm}}{A{\;}_{blank\;517nm}}\right]\times 100$$
where A _sample 517nm_ is the absorbance of the sample with ethanol solution of DPPH at 517 nm after incubation, A _control 517nm_ is the absorbance of the sample with ethanol solution at 517 nm after incubation, A _blank 517nm_ is the absorbance of deionized water and ethanol solution of DPPH at 517 nm after incubation.

#### 2.4.2. Superoxide-Radical Scavenging Activity Assay

We measured the superoxide radical scavenging activity according to some methods of research [25]. O_2_•− was generated in a nonenzymatic phenazine methosulfate–nicotinamide adenine dinucleotide (PMS/NADH) system. The method is as follows: The Tris–HCl buffer (16 mM, pH 8.0) was prepared first and then the solution of NADH (365.7 μg mL^−1^), nitro blue tetrazolium (NBT 245.3 μg mL^−1),^ and PMS (18.38 μg mL^−1^) was prepared in Tris–HCl buffer (pH = 8.0). Chitosan and its derivatives with 1 mL concentration of 0.1, 0.2, 0.4, 0.8, and 1.6 mg/mL were added to the test tube (use isovolumetric distilled water as contrast). The reaction mixture, a total volume 3.0 mL, containing 1 mL the sample solution (0.1, 0.2, 0.4, 0.8, and 1.6 mg/mL), was incubated with Tris–HCl buffer (0.5 mL), NADH (0.5 mL), NBT (0.5 mL), and PMS (0.5 mL) at 25 °C for 30 min in a dark place. Control group: 0.5 mL NADH took the place of the 0.5 mL Tris–HCl buffer. The ultraviolet absorbance of the mixture at 560 nm was determined. Vc was used as a positive control. Three replicates for each sample were tested and the scavenging rate of superoxide anions was calculated according to the following equation:(8)$$Scavenging\;effect\;\left(\%\right)=\left[1-\frac{A_{sample\;560\;nm}-A_{control\;560\;nm}}{A_{blank\;560\;nm}}\right]\times 100$$
where A _sample 560nm_ is the absorbance of the sample with configured test solution at 560 nm after incubation, A _control 560nm_ is the sample with configured test solution that the nicotinamide adenine dinucleotide reduced (NADH) solution was replaced the Tris-HCl buffer at 560 nm after incubation, A _blank 560nm_ is the absorbance of deionized water and configured test solution at 560 nm after incubation.

#### 2.4.3. Hydroxyl-Radical Scavenging Activity Assay

The formation of •OH from Fenton reagents was quantified with H_2_O_2_ oxidation. We measured the hydroxyl radical scavenging activity, according to Guo and Liu [26]. The method was as follows: Tthe phosphate-buffered saline (pH = 7.4) was prepared first. Then a solution of H_2_O_2_ (3%) and safranine T (360 μg m L^−1^) were prepared in phosphate-buffered saline (pH = 7.4). A solution of EDTA–Fe^2+^(2 mM) was prepared in water. Chitosan and its derivatives with 1 mL concentration of 0.1, 0.2, 0.4, 0.8, and 1.6 mg/mL were added to the test tube (use isovolumetric distilled water as contrast), respectively. The reaction mixture, a total volume 4.5 mL, containing 1 mL sample solution with concentration of 0.1, 0.2, 0.4, 0.8, and 1.6 mg/mL, was incubated with EDTA–Fe^2+^solution (0.5 mL), phosphate-buffered saline (pH = 7.4, 1 mL), safranine T (1 mL), and H_2_O_2_(1 mL) at 37°C for 30 min. Control group: 1 mL H_2_O_2_ took the place of the 1 mL phosphate-buffered saline. The ultraviolet absorbance at 520 nm of the mixed solution was determined and recorded, and Vc was used as a positive control. Three replicates for each sample were tested. The scavenging rate of hydroxyl radicals was calculated according to the following equation:(9)$$Scavenging\;effect\;\left(\%\right)=\frac{A_{sample\;520\;nm}-A_{blank\;520\;nm}}{A_{control\;520\;nm}-A_{blank\;520\;nm}}\times 100$$
where A _sample 517nm_ is the absorbance of the sample with configured test solution at 517 nm after incubation, A _control 517nm_ is the deionized water with configured test solution without hydrogen peroxide solution at 517 nm after incubation, A _blank 517nm_ is the absorbance of deionized water and configured test solution at 517 nm after incubation.

The microplate reader absorbance of the tested mixture was measured with a DNM-9602G microplate reader (China, pulang new technology co. LTD., Beijing, China). The results were processed by computer programs Excel (Microsoft, Redmond, WA, DC, USA) and Origin 8 (Origin Lab, Northampton, MA, USA) and reported as mean ± SD.

### 2.5. Cytotoxicity Assay

The cells are mouse fibroblasts called “Mouse connective tissue L cell line 929 clone” and they were cultured in RPMI medium with 1% penicillin and streptomycin and 10% fetal bovine serum at 37 °C and maintained under 5% CO_2_ atmosphere. The cell viability of all sample’s derivatives was evaluated by CCK-8 assay in vitro with minor modification. L929 cell lines were seeded into a 96-well plate and incubated for 24 h at 37 °C in a 5% CO_2_ humidified atmosphere. The cells were incubated with increasing concentrations of all sample’s derivatives for 24 h. After 24 h, cell viability was evaluated using CCK-8 by recording the absorbance at 450 nm according to the Elx808 microplate reader which was produced from Biotek. The cells treated with culture medium alone considered as a control and recording cell viability using the following equation [27]:(10)$$Cell\;viability\;\left(\%\right)=\frac{A{\;}_{sample\;450nm}-\;A{\;}_{blank\;450nm}}{A{\;}_{negative\;450nm}-\;A{\;}_{blank\;450nm}}]\times 100$$
where A _sample 450nm_ is the absorbance of the samples (containing cells, CCK-8 solution, and sample solution) at 450 nm, A _blank 450nm_ is the absorbance of the blank (containing RPMI medium and CCK-8 solution) at 450 nm, and A _negative 450nm_ is the absorbance of the negative (containing cells and CCK-8 solution) at 450 nm.

## 3. Results and Discussion

The synthetic strategy of the novel thiourea salt-functionalized chitosan derivatives is shown in Scheme 1. Firstly, *N, O-*carboxymethyl chitosan was synthesized in a solution containing chloroacetic acid and isopropanol solution at 50 °C. Then, *N, O-*carboxymethyl chitosan derivatives containing thiourea salts were synthesized by combining carboxymethyl chitosan with thiourea salt. Each step of the synthesis was followed by ^13^C NMR and elemental analyses. The FT-IR spectra of chitosan and chitosan derivatives are shown in Section 3.1.2. The ^13^C NMR of chitosan and chitosan derivatives are shown in Section 3.1.3. Meanwhile, the yields and the degrees of substitution of chitosan derivatives are shown in Table 1.

### 3.1. Structure of CS, N, O-CMCS, and Carboxymethyl Chitosan Containing Thiourea Salts

#### 3.1.1. Elemental Analyses

Table 1 shows the yields and the degrees of substitution of chitosan derivatives. The changes in W_C/N_ or W_C/S_ indicate the target functional groups grafting on chitosan successfully. The data in the table shows Carboxymethyl chitosan has a high degree of substitution and yield. The degrees of substitution of the carboxymethyl chitosan derivatives containing thiourea salts are different by about 8%. These data can preliminarily prove that the thiourea salt groups are incorporated into the chitosan structure, but further confirmation is needed.

#### 3.1.2. Infrared Spectroscopy

The FTIR spectrum for chitosan, *N, O-*CMCS, 4TMCMCS, TCMCS, 4,4DCMCS, and 4ACMCS were recorded (As shown in Figure 2). For chitosan, the peaks appear at 3373, 2880, 1597, and 1073 cm^–1^. The peak appears at approximately 3373 cm^−1^, indicating the presence of the hydroxyl groups and amino groups of chitosan. The weak peak at 2880 cm^−1^ assigns to the C–H stretching vibration. The peak at 1597 cm^–1^ can be attributed to amino groups of chitosan. In the fingerprint region (1200–990 cm^−1^), the characteristic band appears at 1073 cm^−1^, which is attributed to the stretching vibrations of the glucosamine ring C–O stretch. Compared with chitosan, the carbonyl of carboxymethyl chitosan exists in the form of sodium salt, and the infrared spectrum has not changed. In *N, O-*CMCS, the peaks at 1613 cm^−1^ were assigned to carbonyl groups of –COONa owing to carboxymethylation. Carboxymethyl sodium will be formed as carboxylic acid through the addition of a little bit of hydrochloric acid, and the peak at 1725 cm^−1^ will appear [28]. The FT-IR spectrum of the compound showed absorption bands in 1725 cm^−1.^ This indirectly proved that carboxymethyl chitosan was synthesized successfully. The new peaks of thiourea salts could be clearly observed at 1230 cm^−1^ for 4TCMCS, 1230 cm^−1^ for TCMCS, 1245 cm^−1^ for 4,4DCMCS, and 1246 cm^–1^ for 4ACMCS, respectively. The spectra of these derivatives can be attributed to stretching vibration of C=S, and the appearance of the proton in thiourea salts at 1245 cm^–1^ further proved the thiourea salt groups were incorporated into the chitosan structure, but further confirmation is needed [29].

#### 3.1.3. NMR Spectra

A short synthetic route was designed to complete the chemical modification of chitosan. *N, O-*carboxymethyl chitosan was selected as an intermediate to combine thiourea salts with chitosan (As shown in Figure 3). Meanwhile, carboxymethylation also helps to increase the solubility of the chitosan derivative. The NMR method is the most effective technique currently to determine the structure of chitosan derivatives. The ^13^C NMR results, obtained for prepared compounds at ambient temperature in D_2_O, are presented in this section. ^13^C NMR spectra indicate that these compounds were synthesized. In the ^13^H NMR spectrum of chitosan, the peak at around 100 ppm was attributed to C–1, peaks at 78 ppm to 70 ppm were attributed to C–3, C–4 and C–5, peaks at 61 ppm and 56 ppm was attributed to C–6 and C–2, respectively. In the ^13^H NMR spectrum of *N, O-*carboxymethyl chitosan, the signals at 178 ppm were attributed to the carbons in carboxymethyl ester bonds, and the signals at 65 ppm were attributed to the carbons in carboxymethyl methylene [30]. For carboxymethyl chitosan containing thiourea salts, the chemical shift of the carbon in the carboxymethyl ester bond is shifted to a higher field, and it appears at 175 ppm. The peak at around 180 ppm was attributed to C=S [31]. Based on these results, it is concluded that chitosan derivatives had been synthesized successfully.

#### 3.1.4. Thermogravimetric and Derivative Thermogravimetric Analysis (TGA/DTG)

Thermal gravimetric analysis of CS and chitosan derivatives were used to investigate the thermal degradation and crystallization of the polymers. Figure 4 shows the TG curves and the corresponding derivate-grams (DTG) of CS and chitosan derivatives. The thermal decomposition of chitosan can be divided into two stages: The chitosan underwent a 3.2% loss of mass between 35 and 102 °C, which resulted from the evaporation of water already within the polymer structure. The second stage appeared between 200–275 °C, and the thermal decomposition rate of chitosan reached the maximum at 248 °C. This mass loss could be attributed to oxidation, combustion, and decomposition of the deacetylated units of CS. The thermal decomposition of *N, O*-CMCS has three stages, and the temperature corresponding to the maximum decomposition rate of *N, O*-CMCS is 170 °C. Compared with the TG curve of chitosan, the thermogravimetry of *N, O*-CMCS is not as good as that of chitosan, which may be due to the introduction of carbonyl groups into the structure of chitosan, which destroys the crystal structure of chitosan. In this study, the thermal decomposition rate of CS, *N, O*-CMCS, 4MTCMCS, TCMCS,4,4DCMCS,4ACMCS reached the maximums of 248 °C, 170 °C, 237.5 °C, 230 °C, 237.5 °C, 227.5 °C, respectively. This poor thermal stability of chitosan derivatives suggested that the groups of substituted groups resulted in the breakage of the hydrogen bond of CS.

### 3.2. Antioxidant Activity

#### 3.2.1. Scavenging Ability of DPPH Radical

DPPH, also known as 1, 1-diphenyl-2-trinitrophenylhydrazine, is a very stable free radical in the nitrogen center. Due to the space barrier of the three benzene rings with resonance stabilization effect, the unpaired electrons on the middle nitrogen atoms cannot play their due electron pairing effect. It is purple when dissolved in anhydrous ethanol, and has the maximum absorption at the wavelength of 517 nm [24,32,33]. When DPPH can be combined or replaced with free radical scavenger to reduce the number of free radicals, the absorbance will be reduced, and the solution color will become lighter, so as to evaluate the ability of scavenging free radicals. Figure 5 shows the curve chart of the DPPH radicals’ scavenging ability of CS and its derivatives at various concentrations. Compared with Vitamin C as a control standard, the scavenging effect increased as the concentration of the polymer samples increased. When the concentration is 1.6 mg/mL, scavenging indices were listed as follows: Vitamin C 100%, chitosan 58.69%, CMCS 36.78%, 4MTCMCS 87%, TCMCS 100%, 4,4DCMCS 100%, and 4ACMCS 100%. Therefore, chitosan derivatives containing thiourea salts exhibited the highest DPPH-radical scavenging ability. In other words, thiourea salt groups are an important factor that influences the scavenging activity against DPPH-radicals.

#### 3.2.2. Scavenging Ability of the Superoxide Radical

NBT, which is a common chromogenic agent with maximum absorption at 560 nm, forms purple fornaman with superoxide anions. The superoxide radical-scavenging activity of chitosan and its related derivatives was tested by their ability to bleach the O_2_•−generated from the PMS-NADH reaction [30]. As shown in Figure 6, Vitamin C used as a positive control, the superoxide-radical scavenging ability of the obtained derivatives had excellent scavenging activity. The maximum of 100%, 100%, and 55.88% inhibition was observed at a concentration of 1.6 mg/mL of 4ACMCS, 4,4DCMCS and CS. Therefore, the increase of thiourea salt groups in chitosan derivatives promotes the scavenging effect. The results again suggest that carboxymethyl chitosan containing thiourea salts can be considered as efficient antioxidant polymers, and thiourea salt groups play a role in their free radical scavenging ability.

#### 3.2.3. Scavenging Ability of Hydroxyl Radical

Fenton reaction is an important mechanism for the production of hydroxyl radicals in the body. •OH is produced by the EDTANa_2_-Fe^2+^-H_2_O_2_ system, and hydrogen peroxide can be homogenized to produce hydroxyl radicals under the catalytic action of transition metal ions. The reaction formula is as follows:Fe^2+^ + H_2_O_2_ = •OH + OH^–^ + Fe^3+^

Because the chromogenic saffron can be oxidized and faded under the action of hydroxyl radicals, the absorbance changes. The magnitude of the absorbance can be used to characterize the attack results of hydroxyl radicals, and the content of hydroxyl radicals is measured by colorimetry according to the degree of fading [33]. The hydroxyl radical scavenging activity of chitosan and its derivatives was shown in Figure 7. It can be seen that with the increase of concentration, the scavenging effect of chitosan and its derivatives was increased. Generally speaking, the scavenging effect of chitosan was lower than TCMCS and 4TCMCS, but the scavenging hydroxyl radical effects of 4,4DCMCS and ACMCS are not significant. The scavenging effects of its derivatives on hydroxyl radicals are enhanced or weakened, which is related to the type. These need further study.

In this study, three classic free radical-scavenging tests were carried out. Chitosan derivatives containing thiourea salts have higher antioxidant activity than other polysaccharides noted in the literature, such as the oligo-maltose fraction from Polygonum Cillinerve, different phenolic acids grafted onto chitosan and a new polysaccharide from Bletilla striata fibrous roots (Please refer to Appendix A for details) [34,35,36]. Our results clearly demonstrated the following: Firstly, the antioxidant rates of all tested samples increased with the rising concentration, as seen in Figure 5, Figure 6 and Figure 7. Secondly, the synthesized chitosan derivatives with the thiourea salt at the periphery exhibited potent free radical-scavenging activity. Generally, the antioxidant activity is associated with the density of positive charge, so the carboxymethyl chitosan containing thiourea salt with high-density positive charges would attract more single-electron free radicals, which could perform better antioxidant ability than the carboxymethyl chitosan and chitosan. Thirdly, compared with pristine chitosan and other chitosan derivatives, TCMCS was increased only in thiosemicarbazide groups, but exhibited enhanced antioxidant property. Therefore, it was reasonably deduced that the enhanced antioxidant action of TCMCS was absolutely ascribed to the function of those micarbazide groups. The unique structural features of thiosemicarbazide are mainly reflected in the large dipole, conjugated double bonds, and lone pair electrons of nitrogen atoms, which give rise to excellent electron donors and electron delocalization of ammonium salt. Meanwhile, the results showed that TCMCS exhibited stronger antioxidant properties than 4MTCMCS, 4,4DCMCS, and 4ACMCS, which should be attributed to the appearance of NH_2_ group as a stronger hydrogen donor than secondary amino and tertiary amine group [37,38]. In conclusion, it is reasonable to presume that the thiourea salt groups are a significant factor that influenced the antioxidant activity of all samples, and the structure-activity relationship should be confirmed in the future.

### 3.3. Cytotoxicity Analysis

In order to explore the biocompatibility of chitosan and all sample derivatives at different concentrations, the cytotoxicity test was carried out by CCK-8 assay. Figure 8 and Figure 9 showed the result of the growth of L929 cells treated to different degrees of inhibition after 24 h of treatment using chitosan and all sample derivatives. The samples have less cytotoxicity if the morphology of normal cells is spindle or oval. At 1000 μg/mL, the cell viability of chitosan was about 76%. This meant that pristine chitosan was a bit cytotoxic, but when the tested concentration was less than 500 μg/mL, the cell viability of chitosan was greater than 83.38%. When the tested concentration was 500 μg/mL, the cell viability of CS, *N, O*-CMCS, and carboxymethyl chitosan containing thiourea salt was greater than 80%; some cell viabilities could reach to more than 96%. The results indicated that these derivatives had no cytotoxicity at this concentration [27].

## 4. Conclusions

In summary, a series of derivatives of chitosan with thiourea salt groups were synthesized successfully. In addition, the antioxidant activities of CS, *N, O-*CMCS, and carboxymethyl chitosan derivatives containing thiourea salts were determined. Our results suggested that 4TCMCS, TCMCS, 4,4DCMCS, and 4ADCMCS had higher antioxidant activities. The introduction of the thiourea salt is definitely favorable for the enhancement of the antioxidant action and consequently increases the antioxidant activity of chitosan derivatives. In the cytotoxicity assay, carboxymethyl chitosan derivatives that contained thiourea salts of different structures had less cytotoxic effect. In brief, the study suggested that these designed chitosan derivatives have the advantages of chitosan and thiourea salt, including high antioxidant activity and good biocompatibility. Therefore, carboxymethyl chitosan derivatives containing thiourea salts are endowed with antioxidant activity and can be used as a candidate material in plant growth regulators.

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
