# Peer review of "Synthesis, Characterization, and the Antioxidant Activity of Carboxymethyl Chitosan Derivatives Containing Thiourea Salts"

_polymers, 2019, doi:10.3390/polym11111810_

Round 1
Reviewer 1 Report
Manuscript Presents very good research work related to carboxylated Chitosan ant its application as antioxidant. Manuscript going to be interesting for readers but required Minor revision before consideration to publication. Some minor comments are as follows.
(1) Author need to incorporate some interesting structural characterization data in the abstract part of the manuscript.
(2) Authors need to incorporate some crescent references related to the preparation of carboxylated natural materials and their important application; some examples are as follows;
(a) materialstoday Volume, 21, Issue 7, September 2018, Pages 720-748 (b) Biomacromolecules 18 (8), 2333-2342 (c) ACS Sustainable Chemistry & Engineering 6 (3), 3279-3290 (d) Industrial & Engineering Chemistry Research 56 (46), 13885-13893 (e) ACS Sustainable Chem. Eng., 2019, 7 (6), pp 6140–6151 (f) Chemical Communications 49 (78), 8818-8820 (g) Cellulose 24 (12), 5417-5429 (h) Nanoscale, 2014, 6, 7764–7779 (i) ACS Sustainable Chem. Eng. 2018, 6, 2807−2828 (k) ACS Sustainable Chem. Eng. 2016, 4, 2632−2643
(3) cm- need to replace by Cm- (line 94, page 3)
(4) Authors need to provide comparative antioxidant efficiency their material with reference to other reported similar materials in tabulated form to explain superiority of their material.
(5) Correct oC (line 233, page 6).
(6) Author need to correct scheme (line 145, page 4).
(7) Author need include yields in each step of their synthesis.
(8) Author need to improve quality of figure 2 and also need include the solvent in information in the text.
(9) Correct line 263 and 265.
(10) Why there is maximum antioxidant activity towards TCMCS? Author need to provide suitable explaination.
(11) What pH conditions authors have used during the synthesis of Chitosan-thiourea?
(12) Authors need to include TGA or powder X-ray studies to provide better characterization data for the chitosan and derivative samples.
Author Response
Dear reviewer,
Thank you for your comments concerning our manuscript entitled “Synthesis, characterization, and the antioxidant activity of carboxymethyl chitosan derivatives containing thiourea salts”. Those comments are all valuable and very helpful for revising and improving our paper. We have studied comments carefully and have made corrections which we hope meet with approval. The main corrections in the manuscript was as following:
Author need to incorporate some interesting structural characterization data in the abstract part of the manuscript.
Answer: Thank you for your kind suggestions and according to your recommendation, the abstract is modified as follows:
A new class of chitosan derivatives possessing thiourea salts were synthesized to improve the solubility and the antioxidant activity of chitosan. Firstly, chitosan was modified to carboxymethyl chitosan, combined carboxymethyl chitosan with thiourea salts that have with different structures to form new chitosan derivatives. The chitosan and chitosan derivatives were characterized by FT-IR, 13C NMR, TGA, and elemental analyses. The new peaks of thiourea salts could be clearly observed at about 1240cm-1 in the IR spectra and the peak of C=S was clearly observed at around 180 ppm in the 13C NMR. IR spectra and 13C NMR of the structural units of these polymers validated the chitosan derivatives possessing thiourea salts were successfully synthesized. Their antioxidant properties were tested, including DPPH-radical scavenging ability, superoxide-radical scavenging ability, hydroxyl-radical scavenging ability. Our results suggested the increased of thiourea salt groups in chitosan derivatives promotes the scavenging effect. The scavenging activity of 4TMCMCS, TMCMCS. 4,4DCMCS, and 4ACMCS against DPPH-radical and superoxide-radical were more than 90% at 1.6 mg/mL, respectively. In the cytotoxicity assay, no cytotoxicity was observed for the L929 cells with chitosan and its derivatives at all the testing concentrations. These results demonstrated that the combination of chitosan and thiourea salt groups improved the antioxidant activity of chitosan obviously, and the antioxidants or free radical scavengers based on natural polymers and thiourea salts showed potential applications.
(2) Authors need to incorporate some crescent references related to the preparation of carboxylated natural materials and their important application; some examples are as follows;
(a) materialstoday Volume, 21, Issue 7, September 2018, Pages 720-748 (b) Bio macromolecules 18 (8), 2333-2342 (c) ACS Sustainable Chemistry & Engineering 6 (3), 3279-3290 (d) Industrial & Engineering Chemistry Research 56 (46), 13885-13893 (e) ACS Sustainable Chem. Eng., 2019, 7 (6) , pp 6140–6151 (f) Chemical Communications 49 (78), 8818-8820 (g) Cellulose 24 (12), 5417-5429 (h) Nanoscale, 2014, 6, 7764–7779 (i) ACS Sustainable Chem. Eng. 2018, 6, 2807−2828 (k) ACS Sustainable Chem. Eng. 2016, 4, 2632−2643
Answer: Thank you for your kind suggestions and thank you for providing us with such useful literature, which is very valuable to us, and we have also made an important supplement to the original content.
(3) cm- need to replace by Cm- (line 94, page 3)
Answer: Thank you for your kind suggestions and the Cm-1 has been corrected in the original.
(4) Authors need to provide comparative antioxidant efficiency their material with reference to other reported similar materials in tabulated form to explain superiority of their material.
Answer: Thank you for your kind suggestions and we made the following conclusions by reviewing the relevant literature and comparing other similar antioxidant materials, such as: antioxidant activity is better than other polysaccharides.
Antioxidant activity is better than other polysaccharides.
Types of antioxidant activity |
Sample |
Antioxidant effect |
DPPH radicals’ scavenging ability |
The oligo-maltose fraction from Polygonum Cillinerve [2] |
26.67% (5mg\ml) |
A new polysaccharide from Bletilla striata fibrous roots [1] |
64.67% (5mg\ml) |
|
Different phenolic acids grafted onto chitosan[3] |
80% (2mg\ml) |
|
Carboxymethyl chitosan derivatives containing thiourea salts |
More than 80%, even up to 100% (1.6mg\ml) |
|
Hydroxyl radicals’ scavenging activity |
The oligo-maltose fraction from Polygonum Cillinerve [2] |
91.83% (4.5mg\ml) |
Different phenolic acids grafted onto chitosan [3] |
70% (2mg\ml) |
|
Carboxymethyl chitosan derivatives containing thiourea salts |
More than 55%, even up to 100% (1.6mg\ml) |
|
Superoxide radicals’ scavenging ability |
A new polysaccharide from Bletilla striata fibrous roots [1] |
72.27% (5mg\ml) |
Different phenolic acids grafted onto chitosan [3] |
80% (2mg\ml) |
|
Carboxymethyl chitosan derivatives containing thiourea salts |
More than 90%, even up to 100% (1.6mg\ml) |
|
References |
[1]. Z. Chen, Y. Zhao, X.Wei, Structural characterization and antioxidant activity of a new polysaccharide from Bletilla striata fibrous roots, Carbohydrate Polymers, 2019.115362 [2]. Y. Zhou, W. Ma, L. Wang, Y. Fan, Characterization and antioxidant activity of the oligo-maltose fraction from Polygonum Cillinerve, Carbohydrate Polymers, 2019, 12,15 [3]. Y. Wang, M. Xie, G. Ma, F. Pei, The antioxidant and antimicrobial activities of different phenolic acids grafted onto chitosan, Carbohydrate Polymers, 2019.12.01 |
In this study, three classic free radical-scavenging tests were carried out and chitosan derivatives containing thiourea salts have higher antioxidant activity than other polysaccharides through consulting literature materials, such as the oligo-maltose fraction from Polygonum Cillinerve, different phenolic acids grafted onto chitosan and a new polysaccharide from Bletilla striata fibrous roots (Please refer to Supporting Information) [34-36]. our results clearly demonstrated that: firstly, the antioxidant rates of all tested samples increase with the rise of concentration, as it can be seen from Figs 3-5. Secondly, the synthesized chitosan derivatives with the thiourea salt at the periphery exhibited potent free radical-scavenging activity. Generally, the antioxidant activity is associated with the density of positive charge, so the carboxymethyl chitosan containing thiourea salt with high density positive charges would attract more single electron of free radicals, which could perform better antioxidant ability than the carboxymethyl chitosan and chitosan. Thirdly, compared with pristine chitosan and other chitosan derivatives, TCMCS was increased only a thiosemicarbazide groups, but exhibited enhanced antioxidant property. Therefore, it was reasonably deduced that the enhanced antioxidant action of TCMCS was absolutely ascribed to the function of thiosemicarbazide groups. The unique structural features of thiosemicarbazide mainly reflect in the large dipole, conjugated double bonds, and lone pair electrons of nitrogen atoms, which give rise to excellent electron donors and electron delocalization of ammonium salt. Meanwhile, the results showed that TCMCS exhibited the stronger antioxidant properties than 4MTCMCS, 4,4DCMCS, and 4ACMCS which should be attributed to the appearance of NH2 group as a stronger hydrogen donor than secondary amino and tertiary amine group [37, 38]. In conclusion, it is reasonable to presume that the thiourea salt groups should be a significant factor that influenced the antioxidant activity of all samples and the structure-activity relationship would be confirmed in the future.
(5) Correct oC (line 233, page 6).
Answer: Thank you for your kind suggestions, we are very sorry for the error in the input method, and ℃ has been corrected in the original.
(6) Author need to correct scheme (line 145, page 4).
Answer: Thank you for your kind suggestions and the roadmap has changed as shown in the fig.1.
(7) Author need include yields in each step of their synthesis.
Answer: Thank you for your kind suggestions and the yields of products have been marked out in the original text.
Chitosan (1.62 g, 10 mmol) was stirred vigorously in 20 mL of isopropanol-water system(volume ratio:4:1) at 50℃, then sodium hydroxide (1.2 g, 30 mmol) was added. After stirring for 1 h at 50℃, chloroacetic acid (4.72 g, 50 mmol) was dissolved in 6ml isopropanol, isopropyl chloroacetate solution was then dropped into the chitosan isopropyl alcohol solution slowly. The solution was heated at 50℃ and stirred for 4 h. The product was isolated and washed by pouring the reaction solution into absolute ethanol solution (200 mL) with 70%, 80%, 90% content successively. The precipitate was collected by filtration, washed with ethanol and N, O-CMCS was obtained by freezedrying overnight in vaccum. Yields:75.85%.
A certain amount of thiosemicarbazone (0.876 g, 12 mmol), 4-methylthiosemicarbazide (1.26 g, 12 mmol), 4,4-dimethyl-3-thiosemicarbazide (1.428 g, 12 mmol), and 4-amino-3-indolyl-5-mercapto-1,2,4 triazole (1.752 g, 12 mmol ) was dissolved in 1.2 mol/L hydrochloric acid solution respectively. The volume of hydrochloric acid solution is respectively 10 mL, and stirred for 30 min at room temperature. We will get the thiourea salts that we need, N, O-carboxymethyl chitosan (1.2 g, 2.8 mmol) was dissolved in the thiourea salt solution prepared above, mixed with the above-mentioned thiourea salt, and stirred for 12 h at room temperature. Then, the reacted solution was poured into 100 mL of absolute ethyl alcohol to precipitate the product. The precipitate was filtered and washed carefully for three times with absolute ethyl alcohol, and extracted in a Soxhlet apparatus with ethanol 48 h[19, 20] Then the products were dissolved in water and subjected to dialysis with distilled water for 48 h in a dialysis bag with a molecular weight cut off of 100 g/mol (molecular weight cut off). The products were obtained by freeze drying. Yields:4MTCMCS 42.56%, TCMCS 27.71%, 4,4DCMCS 26.52%, and 4ACMCS 20.41%. (Scheme 1.)
(8) Author need to improve quality of figure 2 and also need include the solvent in information in the text.
Answer: Thank you for your kind suggestions and 50 mg of samples were dissolved in 1.0 mL of 99.9% Deuterium Oxide (D2O) for analyses. In the synthesis process we use a lot of reagents, there will inevitably be impurities. We did our best to purify the sample before doing 13C NMR. Such as the products were dissolved in water and subjected to dialysis with distilled water for 48 h in a dialysis bag with a molecular weight cut off of 100 g/mol (molecular weight cut off), and this is already the best 13C NMR spectrum we have made.
(9) Correct line 263 and 265.
Answer: Thank you for your kind suggestions and the line 263 and 265 has been changed to the following paragraph:
The FTIR spectrum for chitosan, N, O-CMCS, 4TMCMCS, TCMCS, 4,4DCMCS, and 4ACMCS were recorded. For chitosan, the peaks appear at 3373, 2880, 1597, and 1073cm−1. The peak appears at approximately 3373 cm −1, indicate the presence of the hydroxyl groups and amino groups of chitosan. The weak peak at 2880cm−1 assigns to the C-H stretching vibration. The peak at 1597cm−1 can be attributed to amino groups of chitosan. In the fingerprint region (1200–990 cm−1), the characteristic band appears at 1073cm−1, which is attributed to the stretching vibrations of the glucosamine ring C-O stretch. Compared with chitosan, the carbonyl of carboxymethyl chitosan exists in the form of sodium salt, and the infrared spectrum has not changed.
(10) Why there is maximum antioxidant activity towards TCMCS? Author need to provide suitable explaination.
Answer: Thank you for your kind suggestions and the hydrogen bond is existed in amino group which belongs to hydrogen donor, the amino group can provide active hydrogen. compared with pristine chitosan and other chitosan derivatives, TCMCS was increased only a thiosemicarbazide groups, but exhibited enhanced antioxidant property. Therefore, it was reasonably deduced that the enhanced antioxidant action of TCMCS was absolutely ascribed to the function of thiosemicarbazide groups. The unique structural features of thiosemicarbazide mainly reflect in the large dipole, conjugated double bonds, and lone pair electrons of nitrogen atoms, which give rise to excellent electron donors and electron delocalization of ammonium salt. Meanwhile, the results showed that TCMCS exhibited the stronger antioxidant properties than 4MTCMCS, 4,4DCMCS, and 4ACMCS which should be attributed to the appearance of NH2 group as a stronger hydrogen donor than secondary amino and tertiary amine group.
.(11) What pH conditions authors have used during the synthesis of Chitosan-thiourea?
Answer: Thank you for your kind suggestions and the pH conditions of the solvent was as follows during the experiment:
Synthesis of N, O- carboxymethyl chitosan: the amino and hydroxyl groups in chitosan can be activated by sodium hydroxide, the reaction took place in a strong base. Synthesis of 4MTCMCS, TCMCS, 4,4DCMCS, and 4ACMCS: the thiourea salt can only be formed under acidic conditions. In the preparation of chitosan derivatives containing thiourea salts, the pH environment of the solvent is approximately acidic. (12) Authors need to include TGA or powder X-ray studies to provide better characterization data for the chitosan and derivative samples.
Answer: Thank you for your kind suggestions and in order to make the data of this article more comprehensive, we have added TGA analysis test. The data analysis is as follows:
Thermal gravimetric analysis of CS and chitosan derivatives had been used to investigate the thermal degradation and crystallization of the polymers. Fig. 3 shows the TG curves and the corresponding derivate-grams (DTG) of CS and chitosan derivatives. The thermal decomposition of chitosan can be divided into two stages: the chitosan underwent a 3.2 % loss of mass between 35 and 102℃, which resulted from evaporation of water already within the polymer structure. The second stage appeared between 200-275℃, and the thermal decomposition rate of chitosan reached the maximum at 248C.This mass loss could be attributed to oxidation, combustion, and decomposition of the deacetylated units of CS. The thermal decomposition of N,O-CMCS has three stages, the temperature corresponding to the maximum decomposition rate of N,O-CMCS is 170℃. Compared with the TG curve of chitosan, the thermogravimetry of N, O-CMCS is not as good as that of chitosan, which may be due to the introduction of carbonyl groups into the structure of chitosan, which destroys the crystal structure of chitosan. In this study, the thermal decomposition rate of CS, N,O-CMCS, 4MTCMCS, TCMCS,4,4DCMCS,4ACMCS reached the maximum was 248℃, 170℃, 237.5℃, 230℃, 237.5℃, 227.5℃, respectively. This poor thermal stability of chitosan derivatives suggested that the groups of substituted groups resulted in the breakage of the hydrogen bond of CS.

Reviewer 2 Report
Presented in reviewed manuscript results seems to be valuable and worth to be published. However language of the manuscript has to be corrected by a organic or polymer chemist fluent in language (some sentences are simply difficult to understand). After this, manuscript should be reviewed once again.
Some specific comments:
In the cytotoxicity test the safe concentration seems to be 500 ug/mL. However the scavenging experiments were performed at 0 – 1mg/mL. What in your opinion is the application concentration. Is it lower than that found in cytotoxicity test??? Comment please this problems Add explanation for all abbreviations when they are used first time e.g. ….Vitamin C (Vc) ect. If it is possible take the 13C NMR spectrum with more scans to decrease noise-to-signal ratio. In Scheme 1 please assign abbreviation to individual structures. It will easier to read all manuscript Please write what cells are cells of L929 line Could you give any data concerning UV instrument?
Author Response
Dear reviewer,
Thank you for your comments concerning our manuscript entitled “Synthesis, characterization, and the antioxidant activity of carboxymethyl chitosan derivatives containing thiourea salts”. Those comments are all valuable and very helpful for revising and improving our paper. We have studied comments carefully and have made corrections which we hope meet with approval. The main corrections in the manuscript was as following:
. In the cytotoxicity test the safe concentration seems to be 500 ug/mL. However the scavenging experiments were performed at 0 – 1mg/mL. What in your opinion is the application concentration. Is it lower than that found in cytotoxicity test??? Comment please these problems.
Answer: Thank you for your kind suggestions and the cell viability of samples were greater, it indicated that these derivatives had no cytotoxicity at this concentration. When the tested concentration was 1000 μg/mL, the cell viability of CS, N, O-CMCS, 4TCMCS, TCMCS,4,4DCMCS, 4ACMCS were 75.99%, 91.97%, 75.25%, 100.14%, 82.59%, 79.18%, all samples showed lower cytotoxicity. Because when the tested concentration was 500 μg/mL The shape of the cell and the concentration is optimal, we used the cell morphology diagram at this concentration. If other concentrations of cell morphology are required, we can also provide them.
Fig. 1 The sample concentration in the cell growth pictures was 0 μg/mL
Fig. 2 The sample concentration in the cell growth pictures was 62.5 μg/mL
Fig. 3 The sample concentration in the cell growth pictures was 125 μg/mL
Fig. 4 The sample concentration in the cell growth pictures was 1000 μg/mL
(2). Add explanation for all abbreviations when they are used first time e.g. ….Vitamin C (Vc) ect.
Answer: Thank you for your kind suggestions and according to your recommendation, we will add explanation for all abbreviations when they are used first time.
(3). If it is possible take the 13C NMR spectrum with more scans to decrease noise-to-signal ratio.
Answer: Thank you for your kind suggestions. In the synthesis process we use a lot of reagents, there will inevitably be impurities. We did our best to purify the sample before doing 13C NMR. Such as the products were dissolved in water and subjected to dialysis with distilled water for 48 h in a dialysis bag with a molecular weight cut off of 100 g/mol (molecular weight cut off), and this is already the best 13C NMR spectrum we have made.

Round 2
Reviewer 2 Report
The authors have taken into account all my comments and now, in my opinion, the manuscript can be published.